# Tumor Bed Boost Radiotherapy in the Conservative Treatment of Breast Cancer: A Review of Intra-Operative Techniques and Outcomes

**DOI:** 10.3390/cancers15164025

**Published:** 2023-08-08

**Authors:** Javier Sanz, Arantxa Eraso, Reyes Ibáñez, Rachel Williams, Manuel Algara

**Affiliations:** 1Radiation Oncology Department, Hospital del Mar Barcelona, 08003 Barcelona, Spain; 2Facultat de Medicina, Campus del Mar, Universitat Pompeu Fabra, 08002 Barcelona, Spain; 3Radiation Oncology Department, Institut Català d’Oncologia, Hospital Trueta, 17007 Girona, Spain; aeraso@iconcologia.net; 4Facultat de Medicina, Campus Centre, Universitat de Girona, 17004 Girona, Spain; 5Radiation Oncology Department, Hospital Universitario Miguel Servet, 50009 Zaragoza, Spain; reyesibanez@gmail.com; 6College of Liberal Arts, Texas A&M University, College Station, TX 77843, USA; texas.raw@gmail.com

**Keywords:** breast cancer, conservative treatment, whole breast irradiation (WBI), tumor bed boost, intraoperative radiotherapy, outcomes

## Abstract

**Simple Summary:**

Radiation therapy is an important part of conservative breast cancer treatment. Boost radiation of the tumor bed at enough doses is often necessary to increase local control of the disease. There are several techniques for administering a boost, including intraoperative radiotherapy. This review presents the main published results of the use of an intraoperative boost with electrons and low-energy X-ray for local control, secondary effects and tolerance, as well as the advantages in specific situations such as neo-adjuvant therapy or in combination with reconstructive surgery procedures.

**Abstract:**

Conservative surgery is the preferred treatment in the management of breast cancer followed by adjuvant whole-breast irradiation. Since the tumor bed is the main site of relapse, boost doses are conveniently administered according to risk factors for local relapse to increase the efficacy of the treatment. The benefit of a radiation boost is well established and it can be performed by several techniques like brachytherapy, external radiation or intraoperative radiotherapy. Greater precision in localizing the tumor cavity, immediacy and increased biological response are the main advantages of intraoperative boost irradiation. This modality of treatment can be performed by means of mobile electron accelerators or low-photon X-ray devices. There is a lot of research and some published series analyzing the results of the use of an intraoperative boost as an adjuvant treatment, after neoadjuvant systemic therapy and in combination with some reconstructive surgeries. This review discusses advantages of intraoperative radiotherapy and presents the main results of a boost in terms of local control, survival, tolerance and cosmesis.

## 1. Introduction

Radical mastectomy was the treatment of choice for breast cancer patients in the 20th century but in the nineteen-seventies it was replaced by breast-conserving surgery followed by adjuvant breast irradiation and this is now the standard option to treat patients with localized breast cancer [1,2]. Conservative treatment reaches similar results in terms of survival as mastectomy, or sometimes better [3].

Historically, breast radiotherapy consisted of whole breast irradiation (WBI) administering 48–50 Gy at a classical fractionation of 1.8–2 Gy doses. These classical treatment schedules can be replaced by daily hypofractionated schedules, administering 15–16 fractions between 2.66 Gy and 2.85 Gy [4,5]. After evidence reached by randomized trials, moderate hypofractionation obtained similar results in terms of local control, tolerance and cosmesis. Recently, shorter treatment, consisting of 5 consecutive doses, has presented the same efficacy results and better tolerance for the patient as a result of a quicker return to normal life [6].

Owing to increasing knowledge of the molecular and genetic profile in breast cancer and the deployment of available techniques, nowadays therapies can be more effectively personalized by adapting doses and volumes to be administered but at the same time increasing the complexity in management and prescriptions. In order to diminish local relapse as much as possible, additional doses to the tumor bed are needed with high precision in treatment administration to the tissue at risk. This review discusses the efficacy of tumor bed boost doses and also the advantages and main results when administered by means of intraoperative radiotherapy (IORT).

## 2. Bed Boost in Breast Cancer

The tumor bed is the main site of relapse after conservative surgery. Pathological analysis of surgical specimens has demonstrated that the remaining tumor cells are localized 15–20 mm around the resection cavity [7]. Several factors are predictive for ipsilateral breast recurrence like age, higher grade, positive or close margins and the presence of intraductal carcinoma. In the 1980s, after demonstrating that WBI increases local control, several studies showed the efficacy of additional doses or a boost to the tumor bed. In the Lyon trial, patients younger than 70 years old received subsequent adjuvant WBI of 50 Gy with and without additional doses of 4 fractions of 2.5 Gy by means of an electron beam. Patients receiving a boost presented lower recurrence rates with increased incidence of telangiectasia but similar cosmesis between the groups after a follow-up of 3 years [8].

In the EORTC 22881 trial, patients with negative margins after lumpectomy were randomized to receive a sequential 16 Gy tumor bed boost versus no additional dose. Patients included in the boost group presented higher local control rates with a local recurrence of 7% at 10 years when compared to 10% in the no-boost group (*p* > 0.0001). All subgroups benefited from the increased dose but this was markedly higher in the younger patients. The boost was associated with increased rates of fibrosis and telangiectasia [9].

After WBI with moderate hypofractionation, a bed boost was administered sequentially but frequently at a classical fractionation of 2 Gy [10]. Some trials analyzed the use of a hypofractionated boost after WBI administering an additional 13.32 Gy in four fractions [11] or 3–6 fractions of 2.7 Gy [12] with favorable results in terms of tolerance and cosmesis. Thanks to the huge improvement in radiation techniques, a simultaneous integrated boost is being implemented in most centers while also utilizing ultra-hypofractionated schedules, allowing further shortening of adjuvant treatment, saving time and costs without compromising clinical outcomes [13]. This strategy has shown dosimetric advantages in terms of target-volume coverage and lower doses to organs at risk [14].

## 3. IORT in Conservative Breast Cancer Treatment

Intraoperative procedures have been implemented in recent decades as a way to increase better management of several cancers, minimizing systemic side effects by limiting the irradiated normal tissue volume and increasing the therapeutic index [15]. Also, as recently published, intraoperative management allows a theragnostic approach by diagnosing and treating at the same time during surgery [16]. Accelerated partial-breast irradiation (APBI) focuses radiation only to the tumor bed with a margin, as recurrences occur more frequently in this area and this allows shorter treatment durations while sparing healthy tissue. Several randomized trials comparing APBI to WBI demonstrated similar tumor control after five years in selected patients [17]. GEC-ESTRO and ASTRO have provided guidelines for the selection of treatment in patients eligible for APBI [18,19]. Many APBI techniques have been developed, including brachytherapy [20], external radiation therapy [21] and IORT [22,23]. This last technique allows an extremely short radiation-treatment time during surgery and decreases hospital visits for adjuvant radiation therapy. Also, direct radiation of the surgical bed is administered and allows better protection of nearby organs at risk. IORT can be performed using mobile electron units (Figure 1a,b) or low-energy X-ray systems (Figure 1c,d).

The technique of utilizing electron accelerators is one of the available options and entails, after tumor removal, remodeling the tumor bed by digitally dissecting until reaching the pectoral fascia, creating space to temporarily place a protective disk that prevents the deepening of the beam. Additionally, an approximation of the lateral margins is performed to expose the at-risk tissue for radiation. The most recent technical developments of IORT using low-energy X-ray devices offer better depth penetration and utilize spherical applicators that adapt to the tumor cavity without remodeling it, with evident advantages. These include a smaller treatment volume, a shorter learning curve and reduced radiation-protection requirements, as they do not require special isolation of the operating room. Both techniques involve only a discreet increase in operative time. See Figure 2 for comparison of positioning and dose distribution between treatment devices. Intraoperative radiotherapy has the significant advantage of greater precision in localizing the tumor cavity, immediacy of treatment that hinders tumor repopulation, and an increased immune response due to the effect of a high radiation dose at the peri-tumoral microenvironment [24]. By contrast, not all patients are suitable for the IORT procedure due to breast size or localization of the tumor in the breast. So, cautious selection of candidates to receive it is an important issue.

IORT was included as an option to perform APBI, emphasizing the need for careful selection of patients to be included outside a clinical trial (“suitable group” in guidelines) [19]. For example, there are some exclusion criteria like size superior to 3 cm, extensive ductal carcinoma in situ, the presence of positive nodes or a basal-like molecular subtype, owing to an increased risk for local relapse. The first experiences of IORT were carried out with mobile accelerators that provided radiation by means of an electron beam. The ELIOT trial included 1305 patients, comparing WBI to APBI with electron IORT. At a twelve-year follow-up, the relapse rate was significantly higher in the IORT group than in the WBI group, and overall survival was not different between the two groups [25,26] (Table 1). The improved patient selection demonstrated better results in terms of disease control [27].

More recently, the TARGIT-A trial, using 50 kV X-rays with Intrabeam^®^, 3451 patients were randomized to receive WBI (1730 patients) or IORT (1721 patients). Wound-related complications were similar between groups, but there was significantly less grade 3 or 4 toxicity related to radiotherapy complications with IORT than with WBI [28]. Also, better cosmesis results were observed through a computer-assisted objective system [29], and overall a better quality of life [30]. With a longer follow-up (median 8.6 years, maximum 18.9 years), no statistically significant differences were found for local-recurrence-free survival (hazard ratio 1.13; 95% CI: 0.91 to 1.41; *p* = 0.28) [31], but there were significantly fewer deaths in the IORT group than in the WBI group (see Table 1), attributable to inferior mortality from cardiovascular causes and other cancers (45 vs. 74 events, HR 0.59, *p* = 0.005) This pragmatic trial also included patients that after surgery, and according to pathologic findings, needed complementary WBI. So, in approximately 25% of cases, IORT was administered as a boost and for this reason this strategy began to be utilized. Recently, the Spanish breast radiotherapy research group (GEORM) has published a national consensus on the use of IORT both for APBI and as a boost [32].

**Table 1 cancers-15-04025-t001:** Randomized studies evaluating IORT as partial-breast irradiation after breast-conservation surgery.

Study	No. Patients	IORT Technique	IORT Dose (Single Fraction, Gy)	Median Follow-Up (Years)	Rate of Local Recurrence with IORT (%)	Rate of Local Recurrence with WBI (%)	Hazard Ratio (95% CI, *p* Value)
ELIOT [26]	1305	Electrons	21	12.4	11	2	4.62 (2.68–7.95, *p* < 0.0001)
TARGIT-A [31]	3451	50 kV X-rays	20	8.6	2.11	0.95	1.13 (0.91–1.41, *p* = 0.28)

## 4. BED Boost with IORT

IORT is also an option for performing a planned anticipated tumor bed boost [33]. This technique allows increased treatment precision by means of better localization of the tumor bed, optimizing target-volume coverage and better sparing of normal breast tissue. Other advantages are the shortening of total treatment time, convenience for patients, and better cosmetic outcomes attributable to reduced skin toxicity. An intraoperative boost has been performed with electrons (Table 2) or low-energy X-rays. At Val d’Aurelle Cancer Institute, a series of 50 patients was evaluated. IORT was performed with a dedicated linear accelerator localized centrally between six operating theaters. Using 6–13 MeV, doses of 9–20 Gy were administered at 90% isodose line. Postoperative treatment consisted of WBI of 50 Gy in 2 Gy fractions by means of a cobalt unit. No immediate complications related to the IORT procedure were observed but 6 patients experienced Grade 2 subcutaneous fibrosis and 2 other patients presented Grade 1 telangiectasia. Overall, cosmesis was good to excellent with no relationship to acute and late effects nor the administered IORT dose. Only fibrosis and breast pain correlated with the quality-of-life scores. The median follow-up was 9 years. Only two local recurrences occurred at 8 and 14 years yielding a 96% local control [34]. In the Salzburg series, patients with stage I or II breast cancer received conservative surgery and different boost strategies. The group of 190 receiving an IORT boost of 9 Gy was compared with a cohort of patients with an external electron boost of 12 Gy. WBI consisted of 51 Gy in 30 × 1.7 Gy fractions. After a median follow-up of 4.25 years, in the IORT group no ipsilateral breast recurrence was observed compared to the external electron boost. The crude disease-free survival rate was 95.8% in IORT compared to 84.6% in the external electron boost group. There were no differences between groups according to clinical, tumor, or adjuvant treatment characteristics. The authors hypothesized that the reduction in local relapse with IORT may be due to the immediacy of the dose given at surgery with a potential impact on overall survival [35].

Fastner et al. published the results of IORT with electrons as a boost in an important multicentric retrospective series of an ISIORT database including 1109 patients. The IORT dose varied from 6 to 15 Gy and WBI was administered in 25–28 fractions up to 50–54 Gy. The mean follow-up was 6 years and a reduction in breast cancer recurrence was observed with only 16 (0.8%) recurrences observed in the ipsilateral breast and an overall survival of 91.4% [39]. Local relapses occurred at a range of 12.5 and 151 months. On multivariate analysis only grade 3 was a significant risk factor related to local relapse (*p* = 0.031). Another important series more recently published by Kaiser et al. [42] analyzes the results of an unselected cohort of 770 stage I–III breast cancer patients of all risk types in terms of local control and survival. After IORT of 10 Gy with electrons, WBI followed administering a median dose of 54 Gy. In this series there was a rate of 4.9% (38 patients) of wound complication and 28 patients needed reoperation mainly due to postoperative bleeding. In 91% of evaluated patients cosmesis was rated as satisfactory (excellent/good). After a median follow-up of 10 years, 21 in-breast recurrences were observed (2.7%) yielding a local control of 97.2%. Sorted by molecular subtypes, 10-year different local control rates were observed in 98.7% in luminal A, 98% in luminal B, 87.9% in HER2þ and 89% in triple negative patients, respectively. No other risk factor showed an influence on the rates of ipsilateral breast recurrence.

In a multicenter Italian study, 797 patients received an electron IORT boost but WBI was administered either by standard fractionation (50 Gy in 25 fractions) or by moderate daily hypofractionation (40.5 or 42.56 Gy in 15–16 fractions). Acute toxicity ≥ Grade 2 occurred in 179 patients but no patients reported late toxicity > Grade 2. Mild fibrosis was present in 42% of cases but only 1 case of telangiectasia occurred, demonstrating the clear absence of late skin reactions. The cosmetic result reported by patients was excellent in 10%, good in 20%, fair in 69% and poor in 0.3% of cases. Differences between WBI schedules were not reported. This series also obtained optimal local control and survival [44]. More interestingly in a recent publication of an HIOB prospective trial, all patients included received electron IORT of 11.1 Gy followed by hypofractionated WBI of 40.5 Gy (15 fractions). In this trial, with 1.119 patients eligible for analysis, the results demonstrated that the tolerance was excellent and acute toxicity was mild with Grade 1 and 2 dermatitis of 80% and 8.7% at the end of WBI and Grade 0, 1 or 2 in 37.3%, 56.2% and 6.4%, respectively, at one month after the end of external irradiation. On late evaluation, pain was present in 34.4% (25.8 Grade 1, 8% Grade 2), breast edema in 25% (22.4% Grade 1, 2.6% Grade 2) and fibrosis in 41% (32.9% Grade 1, 7.3% Grade 2). Overall, grade 3–4 toxicities presented in <1% of cases. Cosmesis was not altered at evaluation [46] after three years rated as satisfactory by patients in 86% of cases. There were no local recurrences resulting in an actuarial local control of 100% after four years. The authors state that IORT seems to be superior compared to published evidence before 2010. A similar series published by Leonardi et al. [46], in 481 patients receiving a 12 Gy IORT dose followed by 13 fractions of 2.85 Gy WBI, concluded that IORT with electrons followed by hypofractionated WBI achieves an excellent local control at the cost of tumor bed fibrosis, emphasizing the need to explore lower IORT doses. There are no extensive published experiences comparing IORT with respect to other boost techniques, although in a German monocenter study, two boost techniques (IORT or intensity-modulated external radiotherapy) were compared and resulted in early less-acute side effects with IORT but similar mid- and long-term chronic toxicities and no significant subjective and objective cosmesis differences between groups [45].

An IORT boost is also administered by means of low-photon mobile treatment units (Table 3). In the series by Blank et al., 197 patients received IORT with low photons. After a median follow-up of 37 months there were 5 local invasive relapses, 1 local ductal carcinoma-in-situ, 1 axillary relapse, 6 secondary cancers, and 11 distant metastases, resulting in a 5-year disease-free survival rate of 81.0% and an overall survival rate of 91.3% [49]. Local relapse-free survival (invasive cancers) at 5 years was 97.0%. The incidence of Grade 1–2 fibrosis was 13.8% and only 2 patients had Grade 3 fibrosis. Other toxicities observed included severe pain (n = 4, 6.9%), retraction (n = 17, 29.3%), or breast edema (n = 1, 1.7%) [49]. In a prospective cohort study, IORT as a boost compared well in terms of tolerance, cosmesis and patient-reported outcomes by a Breast-Q questionnaire with respect to patients receiving IORT alone as APBI [50]. Vaidya reported the results of 299 patients receiving a KV 20 Gy boost followed by external WBI and stated that the treatment was generally well tolerated. In an individualized case-control analysis, the expected 5-year estimate for ipsilateral recurrence was of 1.73% (6 recurrences observed against 11 expected) comparing well with the figures obtained in the boost or no-boost EORTC trial (4.3%) and the START trial (2.8%) [51]. In a series with longer follow-up, in 400 patients receiving IORT with low-energy X-rays as a boost, 15 local recurrences occurred, resulting in a local recurrence rate at 5, 10, and 15 years of 2.0%, 6.6%, and 10.1%, respectively. The most common high-grade side effects were fibrosis (21%) and pain (8.6%). The majority of side effects occurred within the first 3 years [52]. The main publications on low-energy X-rays as a boost are shown in Table 2. Immediacy of IORT might eventually show a beneficial effect on the prevention of distant metastasis but this fact has not been completely demonstrated among the series of studies. It can be related to the prevention of possible residual tumor-cell repopulation between surgery and adjuvant radiotherapy, and also to better oxygenation of the tumor bed during operations as a factor for enhanced biological effectiveness, but this hypothesis deserves deeper research.

The use of targeted intraoperative radiotherapy as a tumor bed boost has also been explored in patients undergoing neo-adjuvant therapy with the aim of assuring conservative treatment in high-risk and local advanced breast cancer as well as an increase in local and distant disease control. In a retrospective series published by Kolberg et al., 116 patients treated by neo-adjuvant chemotherapy were treated by IORT low-energy X-rays boost (61 patients) and compared with 55 previous recent consecutive patients treated with an external boost of 10–16 Gy according to their relapse risk factors. Both groups received 50 Gy as WBI. Also, the clinical characteristics of both groups were well balanced. With a median follow-up of 49 months there were no statistical differences in local relapse, being disease-free or breast cancer mortality, but the five-year estimate of overall survival was significantly better with IORT (2 events in the IORT cohort versus 9 events in external boost cohort; HR 0.19, *p* = 0.016). Both groups achieved similar local control. The clinical data in this series seem to support the hypothesis that the benefit of IORT may not be limited to avoiding a geographic and temporal miss but needs further research in prospective trials [56].

Also, IORT is an interesting option for patients receiving oncoplastic conservative surgery as this surgical modality confers important implications in order to assure optimal treatment of the tumor bed prior to tissue mobilization. IORT allows breast-radiation treatment to be performed without affecting the overlying skin, thus cosmetic outcomes tend to be favorable [57]. In a review publication by Malter et al. a series of 149 patients were included. Patients were treated with IORT as a boost (20 Gy, Intrabeam^®^) during primary oncoplastic breast-conserving surgery, followed by WBI (50 Gy in 25 fractions). Oncoplastic procedures consisted in glandular rotation (n = 109), dermo-glandular rotation (n = 29) or tumor-adapted reduction mammoplasty (n = 11). The treatment was well tolerated with no grade 3 or 4 acute toxicity. Adverse effects following oncoplastic surgery and IORT were infrequent and included wound-healing problems (3%), erythema grade I–II (8%), liponecrosis (2%) and seroma formation (12%). The esthetic outcomes were excellent in more than 90% of patients’ views. At mid-term follow-up there were no local or distant recurrences [58]. So, IORT as a boost in patients submitted to oncoplastic breast surgery can have a huge benefit.

There is only one published randomized trial comparing an IORT boost. A single-institution randomized phase III trial comparing a 10 Gy electron IORT boost plus WBI and WBI plus an external irradiation (EBRT) boost at standard fractionation included 245 patients in two groups with well-balanced characteristics. After a follow-up of 12 years iso-efficacy of both groups was demonstrated as there were 19 local recurrences in both groups (4 true recurrences in the IORT boost group and 5 in the external boost group, the remaining 15 and 14 in other quadrants, respectively). In total, 12 patients developed post-surgical seromas (7 in the IORT group, 5 in the EBRT group) and 7 wound-healing problems occurred (7.8%), 3 of them in the IORT group. Late reactions associated with IORT were not observed, except for two cases of liponecrosis in the treatment area 2 and 3 years after surgery. Cosmetic outcomes were significantly better in the IORT group compared to the EBRT group, and the difference remained significant at any examination, both in the physician’s evaluation and in patients’ evaluations [47].

The TARGIT-B trial is an ongoing randomized superiority trial to test if a tumor bed boost given intraoperatively is superior, and results in better cancer control compared with a tumor bed boost given with external beam radiotherapy [59]. Eligible patients are high risk for local recurrence and will be stratified according to molecular profile. The main outcomes will be local tumor control, the site of relapse, relapse-free survival rates and overall survival. Analysis will include local toxicities and quality of life measured by patient-reported outcome (FACT-B+4 index). With an estimated enrollment of nearly 1800 patients and expected completion in 2022, preliminary results are awaited soon.

## 5. Conclusions

A tumor bed boost diminishes local relapse rates and can be administered after standard or hypofractionated WBI. After lumpectomy, IORT is an amazing way to treat locally due to the immediacy of treatment and precise localization of tissue at risk for local relapse, and has demonstrated its usefulness and efficacy as an APBI technique. Systems with X-rays are more adapted to surgical cavities than electron beam devices. Also, IORT is one of several available techniques to perform a tumor bed boost. It obtains local control rates ranging from 89% to 100%.

IORT as a boost results in a low rate of postsurgical complications. It is well tolerated in terms of acute side effects. On the other hand, chronic toxicities are present in near to one-third of cases at long follow-up. Due to reduced skin toxicity, cosmesis is generally good or excellent. IORT is of high interest in subgroups of patients with high-risk features compromising disease control. Immediacy of treatment could be responsible for the impact on local control and survival. Also, IORT as a boost has been demonstrated to be useful in patients receiving neo-adjuvant treatment and also in patients receiving oncoplastic procedures, as IORT can be performed prior to tissue mobilization.

There is only one published randomized trial of IORT as a boost with electron beam devices and one ongoing randomized trial analyzing the efficacy of low-energy X-ray units compared to an external radiation boost. In the future, deeper knowledge of biological and microenvironmental local effects of IORT will clarify the convenience of this technique in the treatment of breast cancer after conservative surgery especially in the presence of high-risk features.

## Figures and Tables

**Figure 1 cancers-15-04025-f001:**
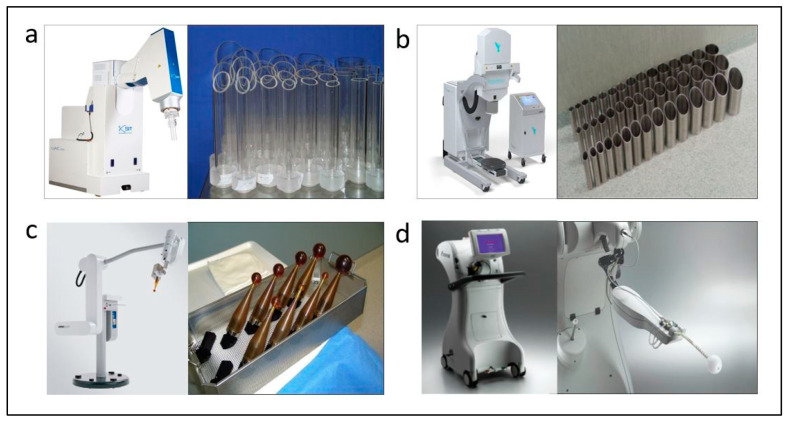
Intraoperative radiotherapy systems and appplicators. In the upper section electron accelerators are shown: Liac^®^, SIT, Vicenza, Italy (**a**) and Mobetron^®^, IntraOp, Sunnyvale, CA, USA (**b**), respectively. At the bottom the systems utilizing low-energy X-rays are shown: Intrabeam^®^, Carl Zeiss, Oberkochen, Germany (**c**) and XoftAxxent^®^, Xoft iCAd Inc., San José, CA, USA (**d**) and spherical applicators.

**Figure 2 cancers-15-04025-f002:**
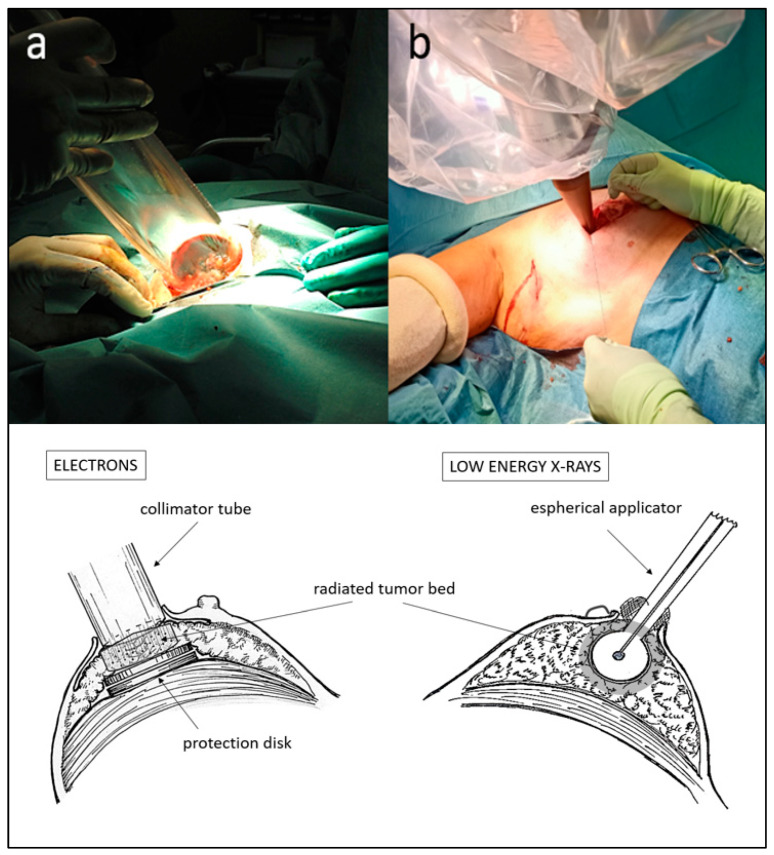
Positioning of IORT applicators for electron accelerators (**a**) and for low-energy X-rays (**b**) after lumpectomy. Schematic dosimetric distributions at the tumor bed in treatment position for each system are shown in the lower half of the figure.

**Table 2 cancers-15-04025-t002:** Main published studies analyzing the use of electron IORT as a boost with doses at surgery and postoperative WBI with several fractionations. (IORT: intraoperative radiotherapy; EBRT: external breast radiotherapy).

Author/Year	Number of Patients	IORT Dose (Gy)	WBI Total Dose/ Number of Fractions	Follow-Up (Years)	Local Control (%)	Overall Survival (%)	Toxicities
			Retrospective				
Batlle/1997 [36]	51	10	45/25	>2	100	NR	13.3% grade 2 fibrosis
Merrick/2003 [37]	21	10–15	45–50/25	5.9	100	90.5	NR
Lemanski/2006 [34]	50	9–20	50/25	9	96	94	12% grade 2 fibrosis
Reitsamer/2006 [35]	190 (IORT) 118 (EBRT)	9 (IORT) 12 (EBRT)	51–56/28	4.25 (IORT) 6.75 (EBRT)	100 (IORT), 95.7 (EBRT)	NR	NR
Ivaldi/2008 [38]	204	13.3	37.05/13	8.9	100		98.2% grade ≤ 2 late skin toxicity
Fastner/2013 (ISIORT registry) [39]	1109	6–15	50–54/25–28	6	99.2	91.4	NR
Fastner/2015 [40]	83 (IORT) 26 (EBRT)	9 (IORT) 12 (EBRT)	51–57/25–28	5 (IORT) 5.6 (EBRT)	98.5 (IORT) 88.1 (EBRT)	86.4 (IORT) 92 (EBRT)	NR
Fastner/2016 [41]	71	7–12	54/27	8	89	75	NR
Kaiser/2018 [42]	770	10	54/24–28	10	97.2	85.7	NR
Machiels/2020 [43]	763	9	40/15 or 50/25	5.2	98.4	97.2	3.5% postoperative complications
Ciabattoni/2022 [44]	797	9–12	40.5/15 or 42.56/16	5	99.25	98.6	22.5% ≥ grade 2 acute toxicity
Schumacher/2022 [45]	76 (IORT) 76 (EBRT)	9 (IORT) 8.4 (SIB EBRT)	50.4/28 or 50/25	7.92	100 (IORT) 98.7 (EBRT)	92 (IORT) 97.4 (EBRT)	37% grade 1 acute dermatitis 21% grade 1 chronic fibrosis
Leonardi/2022 [46]	481	12	37.05/13	9.6	95.9	96.5	<2% acute grade 3 dermatitis 40.8% moderate/severe fibrosis
Prospective/Randomized							
Ciabattoni/2021 [47]	133 (IORT) 112 (EBRT)	10	50/25	12	95.7 (IORT) 94.7(EBRT)	91.6 (IORT) 94.3 (EBRT)	No late complications
Fastner/2022 (HIOB Trial) [48]	1119	11.1	40.5/15	4.2	98.3	97.9	33% grade 1 fibrosis, 9.9% grade 2 fibrosis

**Table 3 cancers-15-04025-t003:** Selected publications of IORT boost by means of systems of low-energy X-rays followed by standard fractionation or moderate hypofractionation schedules for WBI.

Author/Year	Number of Patients	IORT Dose (Gy)	WBI Total Dose/Number of Fractions	Follow-Up (Years)	Local Control (%)	Overall Survival (%)	Toxicities
Retrospective							
Blank/2010 [49]	197	20	45–50/25	3	96.9	91.35	13.8% chronic dermatitis 34.6% grade 2 fibrosis
Wenz/2010 [53]	154	20	45/25	2.83	98.5	87%	30% grade 2 fibrosis 22% grade 1 fibrosis
Vaidja/2011 [51]	299	20	50/25	5	97.4	NR	NR
Pez/2020 [52]	400	20	46–50/23–25	6.5	96.25	81.8	19% and 21.1% chronic fibrosis at 5 and ≥8 years
Tallet/2020 [54]	240	20	46–50/23–25	4.5	99.6	NR	34% grade 1–2 chronic toxicity
Hochertz/2022 [55]	68	20	40.5/15 (16%) 50.4/28 (84%)	7.6	92.6	86.7	1.4 grade 3 acute dermatitis
Prospective/Randomized							
Li/2022 [50]	49	20	40.5/15	1.5	100	100	36.7% grade 1–2 fibrosis

## Data Availability

Not applicable.

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
