# Peer review of "Tumor Bed Boost Radiotherapy in the Conservative Treatment of Breast Cancer: A Review of Intra-Operative Techniques and Outcomes"

_cancers, 2023, doi:10.3390/cancers15164025_

Round 1

Reviewer 1 Report

In the manuscript entitled "Intraoperative radiotherapy tumor bed boost in conservative

treatment of breast cancer: rationale and results review", the authors discussed the recent works on intraoperative radiotherapy the local treatment of breast cancer in terms of local control, survival, tolerance and cosmesis. The topic is interesting. Therefore I recommend the publication after revision:

1. The author discussed the recent achievements using the intraoperative boost with electrons and low-energy x-ray units in terms of local control, secondary effectes ant tolerance. Why intraoperative radiotherapy is necessary, comparing to that in post-surgery or pre-surgery?

2. Breast cancer can be classified into different subtypes, is intraoperative radiotherapy suitable for all the subtypes?

3. The challenges and limitations on treating breast cancer in clinic should be discussed more.

4. More future perspectives may be discussed on the combination therapies with other methods.

5. References for intraoperative treatment on tumor should be updated including these examples; Chemical Engineering Journal 406 (2021) 126900; Adv. Funct. Mater. 2021, 2101320;

6. Please write out the full term for each abbreviation at its first use.

 7. The language should be revised and typos should be corrected.

Moderate editing of English language required

Author Response

Thanks for reviewing the manuscript

We have modify and corrected our work as follows:

1.- Intraoperative radiation therapy offers the advantages of precision and immediacy of treatment that could contribute to better efficacy results mainly in presence of high risk factor or unfavorable molecular profiles. Also in some cases lower volumen of the boost area can contribute to reduced toxicity to surrounding healthy tissues. There are many other advantages related to the shortening total time of treatment and convenience for patients included betteer cosmetic results attributable to reduced skin toxicity.  We have explained best on text by adding a sentence (Another advantages are the shortening of total treatment time, convenience for patients, and better cosmetic outcomes attributable to reduced skin toxicity.”)

2.- Intraoperative radiation therapy is not suitable for all breast cáncer subtypes when administered as exclusive partial breast irradiation following consensus recomendation.

In section 3, third paragrah, we have included a sentence explaining main criteria for partial breast exclusión also applicable to IORT. (“ For example there are some exclusion criteria like size superior to 3 cm, extensive ductal carcinoma in situ, positive nodes presence or basal-like molecular subtype.”)

3.-We have increased length in last paragraph of the introduction to discuss a bit more challenges in breast radiotherapy. (“Due to increasing knowledge of molecular and genetic profile in breast cancer and the deployment of avalaible techniques, nowadays therapies can be more effectively personalized by adapting doses and volumes to be treated but at the same time increasing complexity in management and prescriptions”)

4 .-In page 8 on first paragaph we have made mention about comparison if IORT with other boost modalities (“There are no extensive published experience comparing IORT respect other boost techniques although in a german monocenter study two boost techniques (IORT and intensity-modulated radiotherapy) were compared and resulted in early less acute side effects with IORT but mid and long term same chronic toxicities and no significant subjective and objective cosmesis differences between groups”).

Also we have ameliorated conclusion section including mention about new biological perpectives of IORT.

5.-  At the onset of section 3 we have explained better the concept  of IORT including the better management both in diagnosis and treatment so first of  the references has been included as suggested (Chemical Engineering Journal 406 (2021) 126900). The second reference I have not been capable to insert in the text in order to not extend so much the length of the paragraph.

  1. We have write full term for each abbreviation at its first use along the manuscript.
  2. The complete manuscript has been revised and corrected in spelling, grammar and typos by a native speaker (Raquel Williams), that has been included as a contributing author. For this reason also we have completed and signed the Authorship Form with the agreeing of all authors.

Reviewer 2 Report

Fairly extensive English editing needed overall.

Title end a bit confusing, not apparent it is a general review – suggest:

‘Tumour Bed Boost Radiotherapy in the conservative treatment of breast cancer: A review of intra-operative techniques and outcomes.’

Abstract

gives a good summary of the contents of the review.

Introduction

The statement ‘conservative treatment reaches better results than mastectomy, also in terms of survival’ (ref 3 is somewhat sweeping as this reference is not an RCT or meta-analysis thereof and only found this for certain sub-groups – suggest tone this sentence down a little as the two are generally considered to have similar outcomes within a patient.

Again, classic fractionations of 48-50 Gy at 1.8 to 2Gy continue to be fairly widespread so have not been replaced by hypofractionation I would say, it is region dependant – hypofractionation is a standard rather than the standard.

Note ‘more protracted’ means longer or drawn out – the meaning implied here appears to be the opposite.

Body of Review

Section 2 bed boost – well described though some English editing required.

Section 3 – well described. Further English editing. For TARGIT-A suggest report HR for reduced CVS mortality as this is important.

Suggest trials involved in Section 2 and 3 should also be tabulated.

Section 3 is repeated, I presume ‘BED boost with IORT’ should be section 4.

Suggest Table 1 should be sorted into retrospective series and RCTs then into year of publication, not just the latter.

Also key toxicities prob should be included in the same or a separate table.

Do any further studies explore the reduction in cardiac deaths seen in TARGIT A?

Any data on RT induced cancers with the various techniques.

Conclusions

This is a solid review of the topic with multiple studies covered well.

However, it is largely descriptive of the studies without a robust attempt to summarize and conclude, this conclusion definitely comes across as an afterthought.

Suggest conclusions should also:

A)      Attempt to summarize the most encouraging IORT techniques based on available data and their pros/cons v other boosts (including differences between patient groups where data guides this)

B)      Describe at least in brief what the key studies are required to advance this field and conclude how to best manage patients

There are relatively extensive spelling errors many being basic which suggest the document was not spell-checked prior to submission plus some syntax errors. However, the broader sentence structures are reasonable

Author Response

Thanks for reviewing our manuscript.

We have modify and corrected the work as follows:

1.- The complete manuscript has been revised and corrected in spelling, grammar and typos by a native speaker (Raquel Williams), that has been included as a contributing author. For this reason also we have completed and signed the Authorship Form with the agreeing of all authors.

 2.-  Title has been modified according to your suggestion, clearly more adjusted to the content.Thank you! So, the new title is Tumor Bed Boost Radiotherapy in the conservative treatment of breast cancer: A review of intra-operative techniques and outcomes.”

3.-  We have moderate tone regarding better survival in conservative treatment,so the text in Introduction has been modified conveniently.

4.- We have reformulated the paragraph on fractionation in breast cancer radiotherapy respecting the widespread of schedules worldwide.

5.- We have corrected the misleading term on extension of ultrahypofractionated radiation “protracted” by the opposite “shorter”.

6.-  In page 6, first paragraph, we have include the events and HR with p-value on non breast cancer mortality atributtable mainly to cardiovascular deaths.

7.- Randomized trials on IORT APBI have been included in new table 1 with the main results of them.

8.- We have corrected the numbering of sections 4 (“Bed boost with IORT”) and 5 (“Conclusions”)

9.- We have corrected tables 2 and 3 by grouping studies by type and then by year of publication.

10.- We have included, when posible, the key acute and chronic toxicities of IORT boost studies in tables 2 and 3.

11.- No other published studies have reported reduction in cardiac deaths as seen in TARGIT-A trials, so this question deserves deeper research.  

12.- There are no data on induced cancers through all the published experience on IORT although it seems to be very rare and progresively reduced after a huge improvement in radiation therapy techniques.

13.- We have re-writen completely the conclusion section trying to be more interpretative on the summarized studies, including pros and cons, ongoing research and future perspectives.

Reviewer 3 Report

An interesting manuscript summarizing information about intraoperative radiotherapy.

The authors recall the uses and possibilities of this often forgotten method.

One of the shortcomings of the manuscript is the lack of emphasis on the limitations of this method. I propose to broaden this aspect, then the manuscript will be more objective and I will be happy to recommend it

I also suggest changing the keywords and summary as they do not reflect the content of the manuscript

Author Response

Thanks for reviewing the manuscript

We have modify and corrected our work as follows:

 1.- We have included mention about limitations of IORT, due to the fact that in several cases for technical reasons, anatomy, etc, is not feasible.

2.- We have revised key words and summary.

3.- The complete manuscript has been revised and corrected in spelling, grammar and typos by a native speaker (Raquel Williams), that has been included as a contributing author. For this reason also we have completed and signed the Authorship Form with the agreeing of all authors.

Round 2

Reviewer 1 Report

Accept in present form

Reviewer 2 Report

The modest issues with review structure have been addressed and the tables are more comprehensive plus the conclusion is a better synthesized summary of the contents.

This is a solid piece of work and useful summary.

relatively frequent small English edits are still required.